# Reproductive Dynamics of Spot Tail Mantis Shrimp (*Squilla mantis*): Insights from the Central Mediterranean Sea

**DOI:** 10.3390/ani14172503

**Published:** 2024-08-28

**Authors:** Sabrina Colella, Alessia Mascoli, Fortunata Donato, Monica Panfili, Alberto Santojanni, Giorgia Gioacchini

**Affiliations:** 1National Research Council (CNR), Institute for Marine Biological Resources and Biotechnology (IRBIM), 60125 Ancona, Italy; sabrina.colella@cnr.it (S.C.); monica.panfili@cnr.it (M.P.); alberto.santojanni@cnr.it (A.S.); 2Laboratory of Developmental and Reproductive Biology, DiSVA, Università Politecnica delle Marche, 60131 Ancona, Italy; giorgia.gioacchini@staff.univpm.it

**Keywords:** *Squilla mantis*, sex ratio, reproductive cycle, maturity scale, histological analysis, L_50_, somatic indices, Mediterranean Sea

## Abstract

**Simple Summary:**

*Squilla mantis*, also known as spot tail mantis shrimp, is a stomatopod crustacean widely distributed in the Mediterranean Sea and the adjacent Atlantic Ocean. In Italy, *S. mantis* is the main crustacean species targeted by commercial catches, and in particular, the Adriatic landings represented about 65% of the total national catches in 2021. Despite the economic, ecological, and nutritional importance of *S. mantis*, studies on reproductive biology are scarce or not updated in the Mediterranean Sea, particularly in the Adriatic Sea. In light of the above, the aim of this study was to describe the reproductive aspects of spot tail mantis shrimp, contributing to enriching the knowledge of *S. mantis* reproductive biology. Therefore, our findings could be considered an important tool for correct fisheries management and sustainable exploitation of this commercially fished resource in the Adriatic Sea.

**Abstract:**

Fisheries management requires improvement in scientific knowledge to ensure sustainable exploitation of important commercial species and population renewal. Within this context, this study focused on the reproductive biology of spot tail mantis shrimp, *Squilla mantis*, in the Central Mediterranean Sea, aiming to understand the reproductive pattern of this species and validate the macroscopic maturity scale through histological analysis. A multi-year sampling was performed from 2016 to 2020 by a commercial fishing fleet in the Northern Central Adriatic Sea (GSA 17), and a total of 2206 individuals were collected. The monthly average value of the total sex ratio of *S. mantis* was 0.524 ± 0.044 (*mean ± SEM*) in favor of females, which dominated the population from September to April. The proposed 5 stage macroscopic maturity scale was validated histologically through histological analysis, confirming synchronous ovarian development. The somatic indexes (GSI and K Fulton) and monthly incidence of macroscopic ovarian maturity stages highlighted a protracted reproductive season from winter to spring (January–May). Although the length-weight relationship showed a similar growth trend between genders, males reached a bigger size in terms of carapace length (C.L.) and dominated the population from 32 mm (C.L.). The macroscopic L50 estimated was 25.94 mm (C.L.).

## 1. Introduction

The spot tail mantis shrimp, *Squilla mantis* (L., 1758), is a stomatopod crustacean belonging to the family of *Squillidae*, ecologically important in tropical and temperate ecosystems around the globe [1]. This species is unambiguously identified among other ones of stomatopods by exhibiting distinctive morphological characters described in the literature, in particular a paired dark circle on the telson [2,3]. It is a gonochoristic and oviparous species: females provide maternal care by brooding the egg mass thanks to the material secreted by the cement glands. Furthermore, females ensure that the brooded embryos are kept clean and oxygenated through continuous circulation of water by using the maxillipeds [4] and brood eggs in tunnels to avoid danger, injury, and predators [5]. 

*S. mantis* is widely distributed in the waters of the Mediterranean and the adjacent Atlantic, where it has been reported in particular in the Gulf of Cadiz, in the Canary Islands, and in Madeira [6], from sub-literal depths greater than 3 m [7] up to approximately 150 m depth [8]. It is a benthic species strongly related to bottom sediments, particularly found at high densities in areas with silty sand and sandy mud, substrates suitable for its burrowing behaviour with U-shaped shelters [6,9,10,11,12]. For this reason, mantis shrimp is common in shallow areas with a wide continental shelf [2] and under the influence of river runoff at the mouth of large Mediterranean rivers such as the Ebro, the Rhone, and the Po [6,13]. In particular, eutrophic shallow waters, weak bathymetric gradient, sandy bottom, and considerable nutrient influx from rivers, primarily from the Po River, make the Northern and Central Adriatic Sea one of the most productive zones within the entire Mediterranean Sea [14] and specially adapt for the burrowing behaviour of *S. mantis*.

The diet of spot tail mantis shrimp is based on crustaceans, molluscs, and benthic fish with high prey variability related to local and temporal availability, reflecting the strong dependence on bottom sediments and the opportunistic behaviour of this species. The characteristic burrowing practice of mantis shrimp contributes to the turnover and oxygenation of soft seabed sediments, playing an important role in marine ecosystems [15].

Although ten stomatopod species are found in the Mediterranean and adjacent Atlantic waters, *S. mantis* is the exploited crustacean species most economically important among Mediterranean countries [11,16,17,18]. It is mainly caught by bottom trawlers, although trammel nets, gill nets, and baited traps are also used [6,12,19,20], targeting prawn and lobster species, or fish species such as soles (*Solea* sp.) and red mullets (*Mullus* sp.) [21]. In particular, in the Northern and Middle Adriatic Sea, important catches of mantis shrimp come from a modified beam trawl called “rapido”, specifically employed for flatfish and scallops’ fishery [21], but also efficient for *S. mantis* [22]. In the Northern and Central Adriatic Sea, particularly in the Geographical Sub Area 17 [23], small pelagic fish are dominant in terms of landings, but the main crustacean species targeted by commercial catches include Norway lobster (*Nephrops norvegicus*) and spot tail mantis shrimp (*Squilla mantis*) [24,25,26].

In the last decade, the Italian production of spot tail mantis shrimps fluctuated from 4751 tonnes in 2012 to 4011 tonnes in 2021, and the Adriatic landings have represented about 65% of the total national catches [27]. 

*S. mantis* and other crustaceans are highly appreciated from a nutritional point of view and play a fundamental role in the Mediterranean diet. The beneficial effects of crustaceans’ consumption on human health are related, among other factors, to the high content of polyunsaturated fatty acids, in particular omega-3 fatty acids, essential for human growth and development and fundamental for the prevention and treatment of coronary artery disease, diabetes, hypertension, and cancer [28,29,30].

Despite the economic, ecological, and nutritional importance of *S. mantis*, studies on reproductive biology are scarce or not updated in the Mediterranean Sea, particularly in the Adriatic Sea. Knowledge on the reproduction of fish and other organisms is crucial to understanding the interaction between population and environment and to ensuring the survival and development of the species. One of the main objectives of fisheries management is to enable sustainable exploitation while maintaining an adequate level of spawning biomass to ensure population renewal. For this reason, the knowledge of reproductive patterns, such as length at maturity and spawning cycle, is essential to quantifying the reproductive capacity at both individual and population levels [31]. 

Given the above, the aim of this study was to investigate the reproductive biology of *S. mantis* in the Northern and Central Adriatic Sea (Geographical Sub Area 17—GSA17). The main aspects of the investigation were to: 

(1) increase the knowledge of the reproductive biology of the *S. mantis* population in the Central Mediterranean Sea in terms of the sex ratio, reproductive cycle, spawning season, and temporal trends of the Gonadosomatic Index (GSI) and Fulton’s Condition Factor (K) somatic indices;

(2) validate the proposed macroscopic maturity scale through histology;

(3) estimate length at reproductive maturity (L_50_).

## 2. Materials and Methods

**Sampling and study area**. A multi-year sampling (2016–2020, except in August due to the fishing ban) of wild spot tail mantis shrimps (*Squilla mantis*) was carried out by the Institute for Marine Biological Resources and Biotechnology of the National Research Council (CNR-IRBIM) of Ancona, in the Northern and Central Adriatic Sea (GSA 17) (Figure 1). 

All specimens used in this study were collected in the context of the Data Collection Framework—Biological Sampling of Commercial Catches (DCF, EU Regulation 2017/1004), operative since 2002. The procedures did not include any animal experimentation, and ethics approval was therefore not necessary, in accordance with the Italian legislation (D.L. 4 of March 2014, n. 26, art. 2).

All samples were obtained from a professional fishing fleet on the landing harbours of Ancona and Chioggia, collected at a bathymetric layer of 0–80 m, although most samples were caught up to 50 m fishing ground depth. After collection, samples were saved on ice and transported to the laboratory for analysis.

**Sex ratio and reproductive pattern**. The sex of individuals was determined macroscopically by the presence (male) or absence (female) of a pair of copulatory organs arising from the base of the third pair of pereiopods corresponding to the 8th thoracic segment [13,17]. Moreover, the milky-white cement glands of adult females are well evident through the cuticle of the 6–8 sternite during the spawning period and for a brief time in the post-spawning period [12].

Females’ macroscopic maturity stages were determined according to a 5-stage model proposed by Colella et al. [32], derived from the modification of a reference scale proposed by Froglia [12]. To avoid a misclassification, the “quiescent” and the “spent” stages were grouped into a single one due to their similarity and the consequent possibility to confuse them by macroscopic approach.

The spawning cycle was investigated by analysing the occurrence of each ovary maturity stage throughout the annual cycle of this species.

The sex ratio was monthly estimated during the study period, and it was expressed by dividing the number of females by the total number of females and males (females/males + females), evaluated by month and by length class.

**Morphometric measurements and biometric relations**. Carapace length (CL, mm, from the eye socket to the posterior margin of the carapace) with callipers (accuracy 0.1 mm) and total weight (W, g) to the nearest 0.1 g (Radwag WLC-6F1/K precision scale) were measured for each *S. mantis* specimen.

Length-weight relationships were expressed for each sex by the following equation:(1)$$W=a\;*\;{CL}^{b}$$
where *W* is the total weight, *CL* is the carapace length, *a* is the intercept, and *b* is the slope of the fitting of the observed data. The exponent *b* is the allometry coefficient reflecting the proportionality between carapace length and weight (*b* < 3 negative allometry, *b* = 3 isometry, *b* > 3 positive allometry) [16,33].

For the female subsamples, collected during 2016, the abdomen was dissected, gonads removed, and weighed to the nearest 0.001 g (Mettler Toledo XP204, analytical precision scale), in order to evaluate the monthly variation of the gonadosomatic index (GSI). In addition, the K Fulton condition factor (K) was estimated according to the following equations:(2)$$\%\;GSI=\frac{GoW(g)}{W(g)}*100$$
(3)$$\%\;K=\frac{W\;(g)}{{CL}^{3}(mm)}*100$$
where *GoW* is the gonad weight, *W* is the total weight, and *CL* is the carapace length.

**Histological analysis**. A subsample of females was used to perform histological analysis to validate and reinforce the macroscopic gonads assessment by a microscopic evaluation of the development throughout the reproductive cycle.

The ovary was removed from 6 individuals per macroscopic gonad stage (N = 30), haphazardly chosen from different sampling and at different sizes.

The middle area of the ovary pieces was preserved in the Dietrich solution (900 mL distilled water, 450 mL 95% ethanol, 150 mL 40% formaldehyde, and 30 mL acetic acid) [34], for at least 60 days, and then embedded in paraplast (Paraplast^®^, Sigma-Aldrich, Burlington, NJ, USA), cut in transverse serial sections (6–7 μm), and mounted on slides. Slides were dehydrated by increasing the alcohol concentration protocol and stained with Harris haematoxylin and eosin [35].

Gonad sections were examined at 10–63X magnification under a light microscope (Leica DM 4000) connected to a digitised computer video system (Leica Application Suite 4.3.0.) through a CCD video camera (Leica DFC 420). The histological stage of gonad maturity was assigned according to a five-stage scale [36].

**Size at Sexual Maturity (L_50_)**. All female specimens subjected to macroscopic ovary stage assessment were used to estimate the size at first maturity (L_50_, the length at which 50% of individuals are sexually mature). The L_50_ was calculated by the following logistic function used in the logistic regression model:(4)$$y=\frac{1}{1+e^{-(a+b*x)}}$$
where *y* is the proportion of mature females at length, *a* (intercept) and *b* (slope) represent the estimated parameters, and *x* is the length of the individual (carapace length of *S. mantis*). L_50_ is the total length at which 50% of the females are mature, and it was computed as L_50_ = −a/b. Specimens in the *1-immature* gonadal stage were classified as immature, and specimens from stages 2 to 5 (*2-early maturation, 3-maturation, 4-ripe, 5-spent*) were classified as mature.

**Data analysis**. All the statistical analyses were performed in an R environment, using R software version 4.0.0 (R Core Team, 2020). The Chi-square goodness of fit test was conducted to determine whether the sex ratio (by month and by carapace length class) differed from the expected value (0.5). The null hypothesis (no difference between the observed and the expected proportions) was tested at *p* < 0.05.

A linear regression analysis on log-transformed data was performed to obtain the seed value of *a* and *b* parameters of the length-weight relationship for male and female populations (considering the carapace length instead of the total length). Then, such starting parameters were used to carry out a non-linear regression analysis on the original data. The t-test was performed to detect statistical significance between the length-weight relationship of two sexes.

Statistical differences in gonadosomatic index (GSI) and condition factor K Furton (K) variations were checked by one-way analysis of variance (ANOVA), followed by a post hoc Tukey’s multiple comparison test. The confidence interval was set at 95% (*p* < 0.05), and results were expressed as mean value ± standard error of the mean (SEM).

The L_50_ estimation was calculated by using the sizeMat library in R software and performing the logistic regression model, and the model’s goodness-of-fit was assessed by McFadden’s pseudo-R2 (ρ^2^) [37]. The statistical significance (*p* < 0.05) of the estimated parameters for L_50_ was tested by the Wald test.

## 3. Results

### 3.1. Sex Ratio and Reproductive Pattern

A total of 2206 specimens of wild spot tail mantis shrimps (*S. mantis*) were sampled in GSA 17 and sexed based on the macroscopic evaluation of the gonad (Figure 2): 1149 were females (52.1%) and 1057 were males (47.9%).

The total sex ratio was biased towards females, significantly different from the expected value of 0.5, as indicated by the Chi-square test of goodness of fit (sex ratio = 0.521; χ^2^ = 3.8368, df = 1, *p* = 0.05).

The carapace length ranged from 19 to 49 mm, as shown in Figure 3, but the length classes of 19 and 42 mm consisted of only one female, and the classes of 43, 44, and 49 mm consisted of only a few males.

Therefore, 20–39 mm length classes can be assumed as a representative range of the total population sampled in this study, since more than 99% of specimens belonged to 20–39 mm classes. For all further analysis, length classes of 19 mm and >39 mm were excluded from the dataset because they are not represented by both sexes. In this range 20–39 mm, the total sex ratio is confirmed to be biassed towards females, significantly different from the expected value (sex ratio = 0.524; χ^2^ = 5.0458, df = 1, *p* = 0.025; Table 1).

Females dominated the population from 21 mm to 31 mm, significantly at 24–28 mm, while males dominated at larger size, from 32 mm, significantly at 33 mm, as clearly shown by Figure 4.

The monthly sex ratio had an average value of 0.524 ± 0.044 (*mean ± SEM*), and the trend reported in Figure 5 highlights the dominance of females in total population from September to April, significantly in February (sex ratio = 0.65, χ^2^ = 34.935, *p* ≤ 0.001), March (sex ratio = 0.69, χ^2^ = 4.829, *p* = 0.028), September (sex ratio = 0.61, χ^2^ = 13.545, *p* ≤ 0.001), and October, while males outnumbered the population on summer, significantly in June (sex ratio = 0.21, χ^2^ = 53.061, *p* ≤ 0.001) and July (sex ratio = 0.34, χ^2^ = 37.131, *p* ≤ 0.001).

A total of 1130 females (19 of the 1149 females collected during the five-year period were excluded due to the poor condition of the gonad) were classified based on the macroscopic maturity stage of the ovary (*1-immature*, *2-early maturation*, *3-maturation*, *4-ripe*, *5-spent*), according to the reference scale illustrated in Table 2.

Figure 6 showed the monthly distribution of the ovarian macroscopic maturity stage, giving information on the reproductive cycle of females. A protracted reproductive season from winter to spring (Dec-May) was observed with spawning peaks from March to May: about 71% of the individuals sampled in this period were in *maturation* and *ripe* stages (stages 3 and 4, respectively), showing the body cavity almost completely filled by ovaries clearly visible through the cuticle dorsally and ventrally at telson level. *Spent* or *quiescent* individuals were mainly observed from June to November (48% of the individuals sampled in this period), although still quiescent individuals have also been observed during the breeding season (January–April) in lower percentages. The highest frequency of spent (stage 5) was observed in September, with 118 *spent* individuals out of 157 specimens sampled in this month. *Immature* females were observed during the entire year, with the highest percentage in October (about 42% of the monthly sample).

### 3.2. Morphometric Measurements and Biometric Relations

The entire population of *S. mantis* collected in the GSA 17 in 2016–2020 (N = 2205) was used to estimate the carapace length-weight relationships by gender. For males, the range of carapace length and weight were 20–43 mm (31.55 ± 0.12 mm *mean ± SEM*) and 10.7–89.9 g (43.24 ± 0.42 g *mean ± SEM*), respectively, while for females the range of carapace length and weight were 19–39 mm (30.35 ± 0.11 mm *mean ± SEM*) and 13.3–92.8 g (40.08 ± 0.37 g *mean ± SEM*), respectively.

As illustrated in Figure 7, males (Figure 7A) reached a larger size than females (Figure 7B) in terms of carapace length, and in both cases a statistically significant (*p* < 0.05) power function regression exists between carapace length and total body weight (Figure 7C). The estimated R^2^ for males and females was 0.74 and 0.79, respectively, indicating a very good fit for both sexes (Figure 7C).

The slope coefficients b of the carapace length-weight relationship models indicate a negative allometry (*b* < 3), both for males (*b* = 2.23) and females (*b* = 2.29), which presented a similar growth trend. In fact, the results of statistical analysis indicated no significant difference between the two sexes (*p* = 0.56).

Within all the samples collected throughout 2016, 252 wild spot tail mantis shrimp females, ranging from 20–39 mm of carapace length, were selected for in-depth investigation on the reproductive cycle and the stock health in the GSA 17. For this purpose, the monthly trend of the gonadosomatic index (GSI) and the condition factor K Fulton (K) were analysed, as shown in Figure 8. Both indexes showed a significant statistical variation during the year (*p* < 0.001).

In particular, the maximum levels of GSI were recorded from January to May, after which they sharply dropped (Figure 8A).

Starting in June, minimum values were maintained throughout the summer and autumn before returning to increase again in December. As shown in Figure 8B, the condition factor K Fulton fluctuated throughout the year, reaching the highest peaks in March and July and the lowest in May. All the statistical differences detected in the monthly trend of GSI and K Fulton, evaluated by the one-way ANOVA and the post hoc Tukey’s test, are shown in Table 3.

### 3.3. Histological Analysis

The macroscopic scale was validated and reinforced by a microscopic evaluation of each stage. Histological analysis of the ovary (Figure 9) was carried out on a subsample of thirty individuals (N = 30), representative of the entire size range and all macroscopic gonadal stages. The five stages of gonadal development that define the maturation process of mantis shrimp are characterised as follows:

#### 3.3.1. Stage I: Immature

The nucleus of the oocyte (Po = Primary oocytes; 30–60 µm) is large, with a single large nucleolus in its interior or with more numerous small perinuclear nucleoli. The oil globules are not visible at this stage. The cytoplasm is weakly basophilic. (Figure 9A).

#### 3.3.2. Stage II: Early Maturation

Oocytes (LPo = Late Primary oocytes; 150–200 µm) contain small lipid vesicles that appear on the periphery of the cytoplasm. These vesicles increase in number and size throughout the lipid vesicle stage. The yolk granules do not occupy the inside of the cytoplasm yet. (Figure 9B).

#### 3.3.3. Stage III: Maturation

Vitellogenic oocytes become acidophilic, and yolk granules increase in number and occupy the cytoplasm completely. The shape of the oocyte becomes more irregular and polygonal (Evo = Early Vitellogenic oocytes; 400–600 µm). The nucleus becomes basophilic. (Figure 9C).

#### 3.3.4. Stage IV: Ripe

Large oocytes (LVo = Late Vitellogenic oocytes and ALVo = Advanced Late Vitellogenic oocytes; 400–650 µm) are present, characterised by rounded shape and cytoplasm completely covered with lipid globules, vesicles, and numerous protein granules. At this stage, it is difficult to discern the nucleus within the cytoplasm, which is packed with acidophilic yolk granules, some of which are partially fused. Additionally, some follicle cells can be observed surrounding the oocyte (Figure 9D).

#### 3.3.5. Stage V: Spent

This stage is easily confused with stage I and the quiescent stage. The gonads are almost empty. Mature oocytes, remaining in the ovary after egg deposition, begin to become atretic (Ao = Atretic oocytes; 140–160 µm). The yolk envelope is not found, and yolk granules in the cytoplasm begin to coalesce from the peripheral region. Postovulatory follicles (Pof) are present in females after spawning and recovery. (Figure 9E).

### 3.4. Size at the First Sexual Maturity (L_50_)

All female specimens subjected to macroscopic ovary stage assessment were used to estimate the size at first maturity (N = 1130, range carapace length 20–39 mm). Figure 10 shows the representative difference between immature (Figure 10A) and mature females (Figure 10B), which allows to separate the population for the L_50_ estimation.

The size at first maturity, estimated on the basis of the macroscopic staging of the ovary, was found to be 25.94 mm (carapace length), as shown in Figure 11. The smallest mature (Stage 4) female was observed in the 24 mm length class, although female in the *early maturation* stage was already observed in the 21 mm size class. The value of McFadden’s pseudo-R^2^ (ρ^2^), equal to 0.41, indicated an extremely good model fit, as values of ρ^2^ between 0.2–0.4 are considered equivalent to values of R^2^ between 0.7–0.9 for a linear function [38]. The Wald test (χ^2^ = 184.0) revealed a statistical significance (*p* < 0.001 ****) of the estimated parameters of the logistic regression (*a* = −13.381; *b* = 0.5158).

## 4. Discussion

Fishery resources are currently threatened by overfishing and continuous environmental changes, and to avoid their overexploitation, it is necessary to adopt precise management measures to preserve stocks and allow sustainable consumption. Since, in fisheries science, the knowledge on the reproductive biology of species represents a fundamental point for resource management and it is well known how environmental parameters can affect reproduction, it becomes extremely important to continuously update scientific knowledge about reproductive aspects, including length at maturity and spawning cycles [39].

Although *Squilla mantis* holds significant economic and ecological importance in the Central Adriatic Sea, studies on its reproductive biology are still scarce and outdated. The more recent report on *Squilla mantis* growth and behaviour in this area dates back to 1996 [12].

In light of the above, the aim of the present study was to evaluate the main reproductive parameters of the North Central Adriatic Sea population of *S. mantis* in order to update current knowledge on the reproductive strategy contributing to the sustainable management of this species. In addition to the main objective, this study intends to implement the analysis of gonadal maturity by validating, for the first time in this species, the proposed macroscopic maturity scale through histological analyses in order to ensure a correct classification of specimens based on their reproductive status.

Indeed, the fishery for *S. mantis* is not specifically regulated: it belongs to the general multi-species trawl fishery practiced on continental shelves in the Mediterranean Sea with seasonal changes of target species, together with the shrimp *Melicertus kerathurus* (Forskål, 1775), the sole *Solea solea* (L.,1758), the cuttlefish *Sepia officinalis* (L., 1758), and the red mullets *Mullus* spp. [6]. The European Regulation 1626/1994 is supported by local management measures to preserve stocks, such as restrictions on fishing time, seasonal closures, and working hours limitation per day.

A first overview of the *S. mantis* stock in GSA 17 was given by the sex ratio analysis, which was biased towards females, significantly different from the expected value of 0.5, both for the whole sample and for the sub-sample in the 20–39 mm carapace length range, resulting in 0.521 or 0.524, respectively. These results are in accordance with other studies in the Mediterranean Sea in which females outnumber males: Ragonese et al. [11] reported a sex ratio mean F/M of 1.33 significantly different from 1.0 in the southern coast of Sicily; Saglam et al. [40] found the male:female ratio equal to 1:1.42 in the Aegean Sea; Mili et al. [17] equal to 1:1.12 in the Gulf of Gabes (Tunisia).

Females outnumbered males in carapace length classes from 21 mm to 31 mm, significantly at 24–28 mm, while males dominated the population at larger sizes, from 32 mm to 33 mm. It is also shown by the length-weight relationship, where males reached a bigger size than females in terms of carapace length. The monthly sex ratio revealed important fluctuation during the year: significant domination of females occurred from September to April, while males prevailed from May to July. Similar results in mantis species, underscoring the sex ratio in favor of males in spring and summer, were obtained in previous studies [5,20,32]. Seasonal changes in sex ratio could be mainly related to population dynamics rather than seasonal changes in the fishing effort, which is constant all year round [41]. In fact, females of mantis shrimp exhibit the characteristic behaviour in spring and early summer of incubating the eggs in burrows and do not leave their lairs between spawning and hatching [6,12,42,43], resulting in a decrease of catches.

Sexual maturity is traditionally assessed by macroscopic assignment based on the visual inspection of gonads. In this study, females’ gonads were classified according to the 5-stages of maturity proposed by Colella et al. [32], derived from the reference scale developed by Froglia [12] and suitably modified.

On the basis of monthly frequency distribution of maturity stages, specimens in maturation and ripe stages dominated the community from winter to spring (December–May), particularly in March–May when they represented the greater part of the population (71%), while spent females are mainly observed from June to November. These results demonstrated a protracted reproductive season, in line with other studies in the South Adriatic Sea GSA 18 [44], the Gulf of Cádiz [13,45], and the eastern Ligurian Sea [6]. Some authors suggest that large individuals spawned earlier than smaller ones, resulting in a prolonged breeding season for the entire population [5,46]. This seasonal pattern (protracted reproductive season and release of larvae during spring-early summer) could be a possible adaptive advantage in relation to the spring-early summer plankton blooms that could represent greater nourishment available [47].

The macroscopic approach based on the visual inspection of gonad morphology is usually linked with a possible bias in gonad maturity stage assessment, more in fish species than in shrimps. For this reason, histological analysis is needed to validate the macroscopic approach, obtain a precise classification of gonad development, and consequently, a correct identification of the spawning period of important commercial species [48,49,50]. Although the development of the ovaries in *S. mantis* and in stomatopods in general undergoes macroscopic transformations that allow correct classification during the reproductive season through visual inspection, histology is still recommended to avoid misclassifications between immature and spent individuals, macroscopically very similar to each other. Furthermore, microscopic inspection is essential to distinguish between females in the quiescent state and spent stage, both grouped in Stage V described in the reference scale of this study.

Up to date, only one study investigated the histological characteristics of *S. mantis* ovaries in the Mediterranean Sea [51], identifying 6 maturity stages, while our investigation intends to histologically validate the proposed updated macroscopic classification in 5 stages (*immature*, *early maturation*, *maturation*, *ripe*, and *spent)*.

Histological evaluation suggests a synchronous development of oocytes in the *S. mantis* ovary since almost all oocytes are at the same cellular development stage and extruded simultaneously when mature, as already observed in the Mediterranean Sea [51], the Atlantic Ocean [13], and in other stomatopods in Japan [52].

According to Vila et al. [13], at the onset of winter, together with the maturation of the ovary, the cement glands begin their activity, highlighting a strong relationship between the histological features of the gonad and the macroscopic appearance of the individual: white thoracic bands became visible in maturing or mature females simultaneously with the oocyte vitellogenesis process. Therefore, the temporal activity of the cement glands could be considered a secondary sexual character useful for establishing an accurate macroscopic gonadal maturity stage of *S. mantis*.

The monthly trend of the gonadosomatic index (GSI) is a suitable indicator to support macroscopic evaluation of the reproductive cycle of mantis shrimp and its protracted spawning season during winter and early spring. Indeed, in accordance with the frequency distribution of gonad maturity stages, the highest average values of female GSI were observed from December to May, with a peak in April, significantly higher than in summer and autumn. In a previous study in the central Adriatic Sea, the peak of ovarian maturity was reported in February and March [12], when up to 80% of the females had ripe ovaries, while a high percentage of female in spent stage were observed from April to September, in agreement with our results. A similar trend was reported in the Gulf of Gabes, where the GSI peak occurred in February [17].

The condition factor is an important indicator of organisms “wellbeing”, and the optimum condition is strictly linked to large energy reserves able to ensure the correct development of individuals during the life cycle’s different phases, in particular growth and reproduction, preserving the abundance and equilibrium of a population. In the present study, the Fulton’s condition factor (K) significantly increased in March, before the GSI peak (April-May), suggesting a certain complementarity in the pattern of these two indices 53. This complementarity could indicate that individuals accumulated necessary resources to produce vitellogenic oocytes in preparation for spawning, probably storing them in the hepatopancreas [53]. However, this complementarity was not consistent in the following months, when K showed slight fluctuations, which could be related to variations in food supply during the seasons and to burrowing behaviour of adult females, also depending on the environmental conditions.

The length-weight relationship is often used to study population characteristics of many crustacean species [21] and contribute to a species’ life history understanding [54], such as changes during growth [55,56].

*S. mantis* exhibited a significant negative allometric growth (*b* < 3) both for males and females, without significant differences between sexes, suggesting that as specimens increase in length, it does not cumulate to an increase in weight, or else the body form of this species did not grow at the same proportion (growth in length is not proportional to weight) [15]. Our results are in line with other studies [15], where mantis females [13,40,57] and males [21] showed negative allometry.

Scanty evaluations referring to the size at first maturity (L_50_) have been reported in the Mediterranean Sea and in particular in Italian areas. According to Froglia [12], females of *S. mantis* reached the first sexual maturity at 27 mm of carapace length (C.L.) in the Adriatic Sea, while our results highlighted that the maturity occurs slightly earlier, at 25.94 mm (C.L.). In previous literature, L_50_ for females was estimated in the range of 20–24 mm CL, determined by the onset of cement gland development [7,42,43]. Mili et al. [17] reported that in the Tunisian Sea the estimated mean size at which 50% of females reached sexual maturity was 147.19 mm of total length (T.L.), and it was not evaluated and referred to as CL; in the Gulf of Cadiz (Eastern Central Atlantic Ocean), L_50_ for females was found to be 23.7 mm C.L. [13].

These differences in size at first sexual maturity for crustaceans could be mainly attributed to the different geographic locations of the studied areas and consequently to different environmental conditions [17], such as the abundance and distribution of local populations, competition for space and food availability [32], and anthropogenic activities such as fishing [49].

## 5. Conclusions

This study provided an updated overview of the reproductive pattern of *S. mantis*, contributing to the existing knowledge on its biology. For the first time, a macroscopic maturity scale was histologically validated to ensure an accurate evaluation of the reproductive cycle of this species in the Central Mediterranean Sea (GSA 17). Histological analysis identified the spot tail mantis shrimp as a synchronous species, consistent with observations in other Mediterranean and Atlantic areas. The monthly distribution of maturity stages and the trend of the gonadosomatic index revealed a protracted spawning period, with significant peaks from March to May. These findings mainly offer valuable insights for the correct management and sustainable exploitation of this important fishery resource. The results largely align with previous research, reinforcing that the reproductive strategy of this species in the North-Central Adriatic region of the Mediterranean Sea has not been significantly impacted by fishing activities or environmental changes. However, studies like this are increasingly crucial for the sustainable management of the resource, as any shifts in L50 or the breeding period could necessitate the implementation of minimum catch size regulations or targeted fishing bans.

## Figures and Tables

**Figure 1 animals-14-02503-f001:**
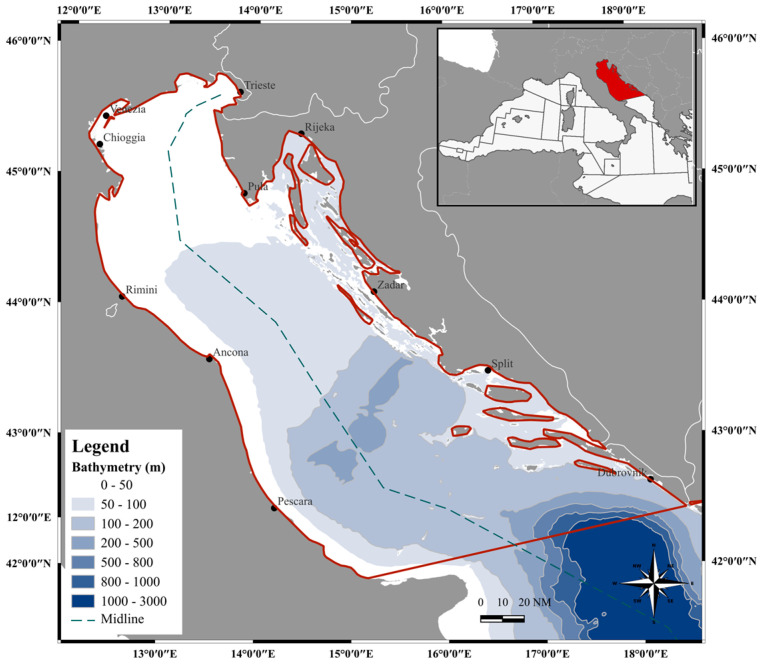
Map of the study area GSA 17, elaborated by I. Costantini (Software: QGIS Development Team, 2024, version 3.36.0. QGIS Geographic Information System. Open Source Geospatial Foundation Project. http://qgis.osgeo.org. accessed on 30 May 2024). Dashed line marks the maritime boundaries between Italy and Croatia.

**Figure 2 animals-14-02503-f002:**
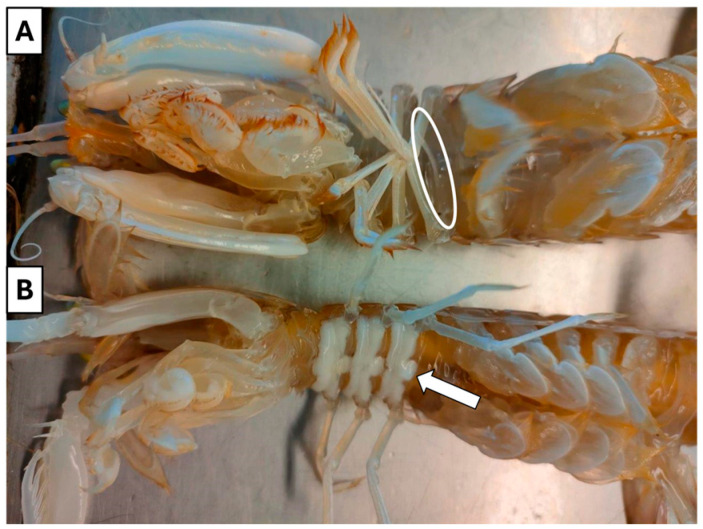
Gender differentiation of wild spot tail mantis shrimps (*S. mantis*) into (**A**) male and (**B**) female. In correspondence with the 8th thoracic segment, the pair of copulatory organs can be appreciated in male (white circle), whereas milky-white colour cement glands are well evident in female through the cuticle of the 6–8 sternite (white arrow).

**Figure 3 animals-14-02503-f003:**
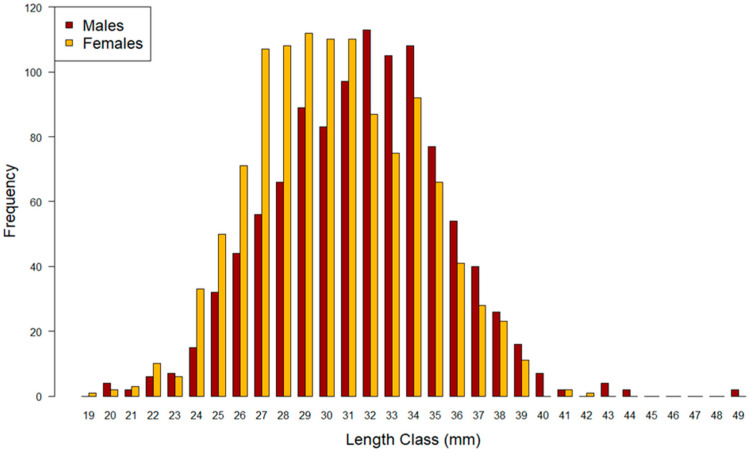
Length frequency distribution of wild spot tail mantis shrimps (*S. mantis*) by sex (N = 2206).

**Figure 4 animals-14-02503-f004:**
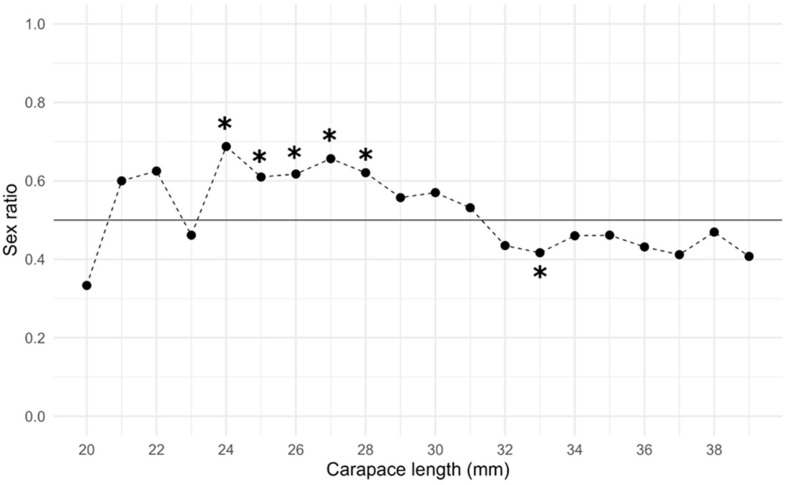
Representation of sex ratio variation of wild spot tail mantis shrimp by carapace length class (N = 2185). The black line represents the expected sex ratio value of 0.5. The sex ratio was estimated by dividing the number of females by the total number of individuals (females/males + females), resulting in greater than 0.5 when females outnumber males and less than 0.5 when males outnumber females. Asterisks indicate a significant difference (*p* < 0.05) in the sex ratio compared to the expected value.

**Figure 5 animals-14-02503-f005:**
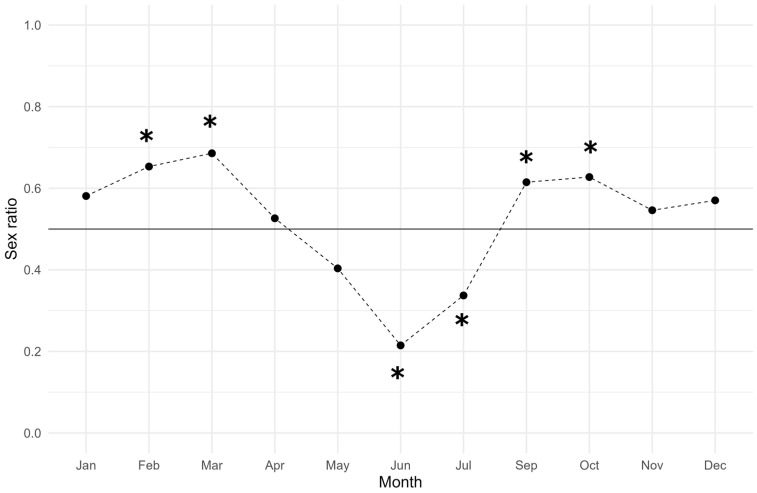
Monthly sex ratio of wild spot tail mantis shrimp throughout five-year period 2016–2020 (N = 2185). The black line represents the expected sex ratio value of 0.5. The sex ratio was estimated by dividing the number of females by the total number of individuals (females/males + females), resulting in greater than 0.5 when females outnumber males and less than 0.5 when males outnumber females. Asterisks indicate a significant difference (*p* < 0.05) in the monthly sex ratio compared to the expected value.

**Figure 6 animals-14-02503-f006:**
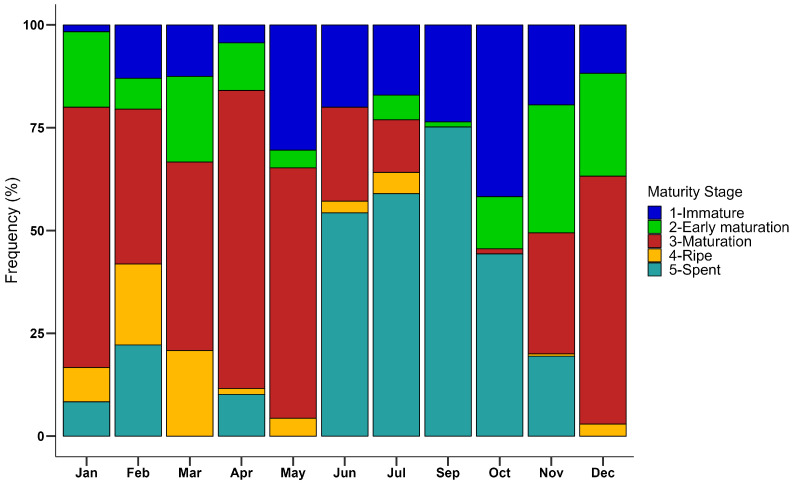
Monthly incidence of different maturity stages of wild spot tail mantis shrimp ovary (*1-immature*, *2-early maturation*, *3-maturation*, *4-ripe*, *5-spent*) assigned using the macroscopic approach (N = 1130). Bars show the relative frequency (%) of each stage by month.

**Figure 7 animals-14-02503-f007:**
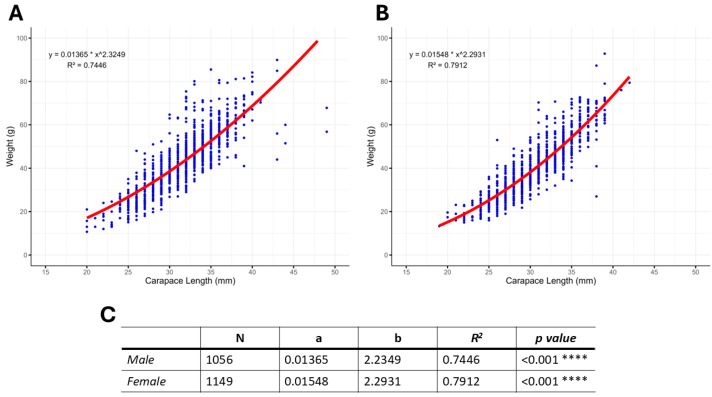
Carapace length-weight relationship in *S. mantis* males (**A**) and females (**B**) collected during the five-year period in the Northern and Central Adriatic Sea (N = 2205). A non-linear regression analysis on data was performed in the R software version 4.0.0 in order to obtain the seed values of *a* and *b*, the R^2^ determinant coefficient, and *p*-value (**C**). Asterisks represent the statistically significant relationship between carapace length and body weigth (*p* < 0.001).

**Figure 8 animals-14-02503-f008:**
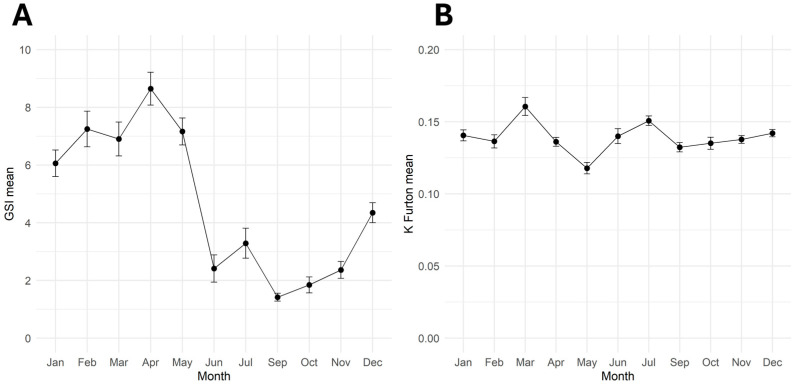
Monthly variation of wild spot tail mantis shrimp female (**A**) GSI mean and (**B**) K mean throughout 2016 (N = 252). Data are reported as *mean ± standard error of mean* (SEM).

**Figure 9 animals-14-02503-f009:**
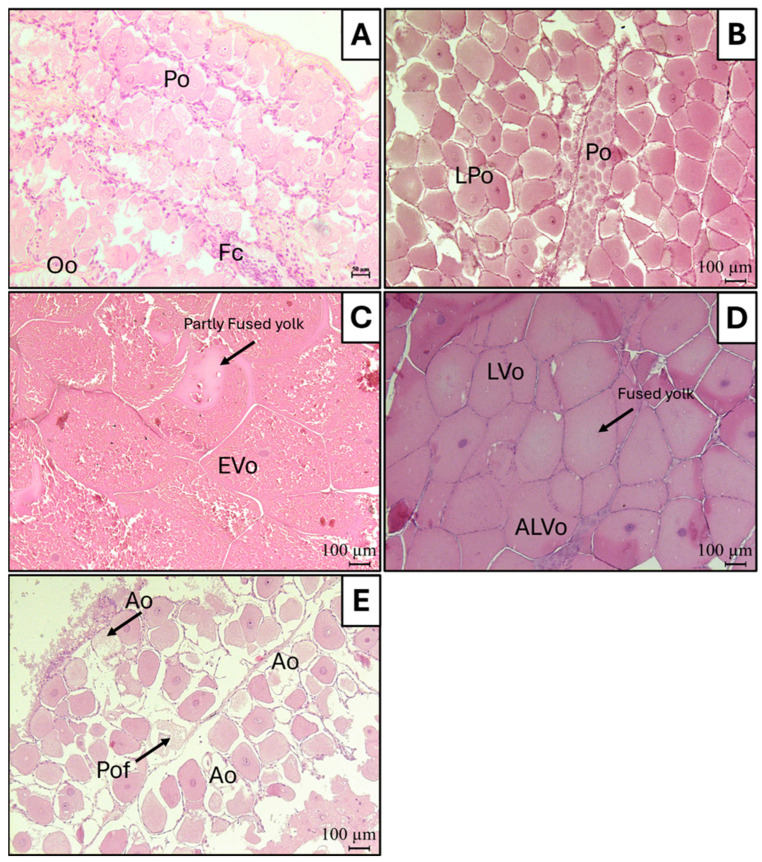
Representative histological photomicrographs of *S. mantis* ovary at different developmental phases: *Stage I Immature* ((**A**), scale bar 50 µm); *Stage II Early maturation* ((**B**), scale bar 100 µm); *Stage III Maturation* ((**C**), scale bar 100 µm); *Stage IV Ripe* ((**D**), scale bar 100 µm); *Stage V Spent* ((**E**), scale bar 100 µm). Abbreviations: Oo = Oogonia; Po = Primary oocytes; LPo = Late Primary oocytes; Evo = Early Vitellogenic oocytes; LVo = Late Vitellogenic oocytes; ALVo = Advanced Late Vitellogenic oocytes; Pof = Post-Ovulatory Follicle; Ao = Atretic oocytes; Fc = Follicle Cells.

**Figure 10 animals-14-02503-f010:**
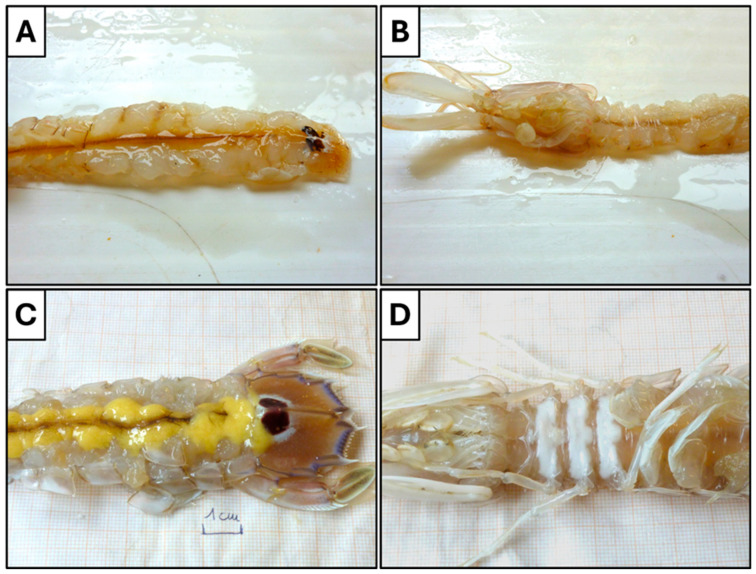
Images of two different ovarian developmental stages of wild spot tail mantis shrimp: stage *1-immature* (**A**,**B**); *stage 3-maturation* (**C**,**D**) in dorsal (**A**,**C**) and ventral (**B**,**D**) view. (Photo: S. Colella).

**Figure 11 animals-14-02503-f011:**
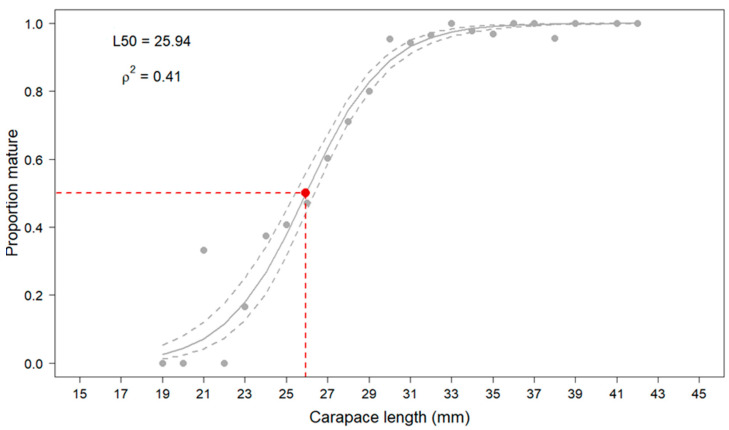
Estimated size at first maturity (L_50_) according to macroscopic classification of wild spot tail mantis shrimp female specimens. The grey dashed lines correspond to the 95% confidence interval. The red dashed line indicates the size of specimens when 50% of female population is mature (size at first maturity). N = 1130.

**Table 1 animals-14-02503-t001:** Summary of the Chi-square test of goodness of fit performed on the sex ratio per carapace length class throughout the five-year period 2016–2020. Bold values indicate a significant difference (*p* < 0.05) in the sex ratio compared to the expected value.

*LENGTH CLASS*	N° of Individuals	N° of Females	N° of Males	Sex Ratio (Females/Males + Females)	χ^2^	*p* Value
20 mm	6	2	4	0.333	0.667	0.4142
21 mm	5	3	2	0.60	0.2	0.6547
22 mm	16	10	6	0.625	1	0.3173
23 mm	13	6	7	0.461	0.077	0.7815
24 mm	48	33	15	0.687	6.75	**0.0093**
25 mm	82	50	32	0.609	3.951	**0.0468**
26 mm	115	71	44	0.617	6.339	**0.0118**
27 mm	163	107	56	0.656	15.957	**<0.001**
28 mm	174	108	66	0.621	10.138	**0.0014**
29 mm	201	112	89	0.557	2.632	0.1047
30 mm	193	110	83	0.570	3.777	0.052
31 mm	207	110	97	0.531	0.816	0.366
32 mm	200	87	113	0.435	3.38	0.066
33 mm	180	75	105	0.417	5.0	**0.025**
34 mm	200	92	108	0.460	1.28	0.258
35 mm	143	66	77	0.462	0.846	0.358
36 mm	95	41	54	0.432	1.779	0.182
37 mm	68	28	40	0.412	2.118	0.146
38 mm	49	23	26	0.469	0.184	0.668
39 mm	27	11	15	0.407	0.926	0.336
**TOTAL**	**2185**	**1145**	**1040**	**0.524**	**5.046**	**0.02469**

**Table 2 animals-14-02503-t002:** Macroscopic descriptions of the phases occurring in the reproductive cycle of wild spot tail mantis shrimp female.

Maturity Stage	Macroscopic Feature
1-*Immature*	Filamentous and hyaline ovaries; sixth to eighth sternites appear hyaline
2-*Early maturation*	Narrow yellow ovaries; sixth to eighth sternites appear whitish
3-*Maturation*	Yellow ovaries, extending up to half of abdomen width, not visible through the cuticle on the ventral side of telson; sixth to eighth sternites appear white
4-*Ripe*	Yellow ovaries, extending over half of abdominal width, visible through cuticle on the ventral side of telson; sixth to eighth sternites appear milky white
5-*Spent*	Filamentous hyaline ovaries with evident brown dots (chromatophores), sometimes still yellow or with a few yellow dots. In this case, the ovaries appear flaccid and shrunken. Sith to eighth sternites appear hyaline or still white.

**Table 3 animals-14-02503-t003:** A summary of the one-way ANOVA followed by post hoc Tukey’s test performed on the monthly variation of GSI and K Fulton Condition Factor. Bold red boxes represent statistically significant differences between months (*p* < 0.05).

GSI	January	February	March	April	May	June	July	September	October	November	December
January		0.7907	0.9658	** 0.0029 **	0.9283	** 0.0220 **	** <0.0001 **	** <0.0001 **	** <0.0001 **	** <0.0001 **	0.2331
February	0.7907		0.9999	0.06599	1.0000	** 0.0006 **	** <0.0001 **	** <0.0001 **	** <0.0001 **	** <0.0001 **	** 0.0035 **
March	0.9658	0.9999		0.2873	0.9999	** 0.0019 **	** <0.0001 **	** <0.0001 **	** <0.0001 **	** <0.0001 **	** 0.0144 **
April	**0.0029**	0.06599	0.2873		0.7183	** <0.0001 **	** <0.0001 **	** <0.0001 **	** <0.0001 **	** <0.0001 **	** <0.0001 **
May	0.9283	1.0000	0.9999	0.7183		** 0.0020 **	** <0.0001 **	** <0.0001 **	** <0.0001 **	** <0.0001 **	** 0.0204 **
June	** 0.0220 **	** 0.0006 **	0.9999	** <0.0001 **	** 0.0020 **		0.9988	0.9979	0.9999	1.0000	0.7759
July	** <0.0001 **	** <0.0001 **	** <0.0001 **	** <0.0001 **	** <0.0001 **	0.9988		0.1423	0.4762	0.9041	0.8353
September	** <0.0001 **	** <0.0001 **	** <0.0001 **	** <0.0001 **	** <0.0001 **	0.9979	0.1423		0.9999	0.9612	** 0.0037 **
October	** <0.0001 **	** <0.0001 **	** <0.0001 **	** <0.0001 **	** <0.0001 **	0.9999	0.4762	0.9999		0.9996	** 0.0249 **
November	** <0.0001 **	** <0.0001 **	** <0.0001 **	** <0.0001 **	** <0.0001 **	1.0000	0.9041	0.9612	0.9996		0.1251
December	0.2331	** 0.0035 **	** 0.0144 **	** <0.0001 **	** 0.0204 **	0.7759	0.8353	** 0.0037 **	** 0.0249 **	0.1251	
**K Fulton**											
January		0.9993	** 0.0100 **	0.9989	** 0.0122 **	1.0000	0.5076	0.9042	0.9953	0.9999	1.0000
February	0.9993		** 0.0011 **	1.0000	0.1256	0.9999	0.1268	0.9997	1.0000	1.0000	0.9949
March	** 0.0100 **	** 0.0011 **		** 0.0011 **	** <0.0001 **	0.4173	0.7931	**0.0001**	** 0.0010 **	** 0.0022 **	0.0589
April	0.9989	1.0000	** 0.0011 **		0.1474	0.9999	0.1204	0.9998	1.0000	0.9999	0.9930
May	** 0.0122 **	0.1256	** <0.0001 **	0.1474		0.3860	** <0.0001 **	0.4936	0.2451	0.0650	** 0.0112 **
June	1.0000	0.9999	0.4173	0.9999	0.3860		0.9719	0.9986	0.9999	1.0000	1.0000
July	0.5076	0.1268	0.7931	0.1204	** <0.0001 **	0.9719		** 0.0171 **	0.0989	0.2049	0.8431
September	0.9042	0.9997	** 0.0001 **	0.9998	0.4936	0.9986	** 0.0171 **		0.9999	0.9963	0.8367
October	0.9953	1.0000	** 0.0010 **	1.0000	0.2450	0.9999	0.0989	0.9999		0.9999	0.9819
November	0.9999	1.0000	** 0.0022 **	0.9999	0.0650	1.0000	0.2049	0.9963	0.9999		0.9993
December	1.0000	0.9949	0.0589	0.9930	** <0.0001 **	1.0000	0.8431	0.8367	0.9819	0.9993	

## Data Availability

On request due to restrictions, e.g., privacy or ethical.

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
