# Peer review of "Reproductive Dynamics of Spot Tail Mantis Shrimp (Squilla mantis): Insights from the Central Mediterranean Sea"

_animals, 2024, doi:10.3390/ani14172503_

Round 1

Reviewer 1 Report

Comments and Suggestions for Authors

Overall well written manuscript.

I would advise you to include the range of oocyte sizes  (ideally oocyte size frequency distribution) for each reproductive stage (where possible).

For sex ratio did you investigate the effect of sampling depth?

Specific comments

Ln 237 in Table 1, correct No of fish with No of individuals.

In the same table either use * for significant difference and ns for non significance or use p values.

Ln 267 Replace Dic with Dec

Ln 297 replace p value with <0.001 and indicate this with 3 asterisks. 

Ln 300 you refer to log-log transformed data, this would result in a linear relationship, on the contrary you express length weight as a curvilinear (power) relationship.

Ln 378 in the results of Figure 11 you could also add for the estimated L50 what is the corresponding weight.

Comments on the Quality of English Language

Quality of English language is fine only minor errors.

Reviewer 2 Report

Comments and Suggestions for Authors

++Simple Summary:

Ensure consistent formatting of the species name, Squilla mantis, in italics throughout the article.

++ Abstract

The technical terms used (e.g., macroscopic maturity scale, histological analysis, somatic indexes, GSI, K Fulton) are suitable for the intended scientific audience. However, defining acronyms such as Gonadosomatic index (GSI) and Fulton’s Condition Factor (K) upon their first mention would be beneficial.

The abstract presents key findings, such as the sex ratio, length-weight relationship, and reproductive season, which are important for understanding the study's outcomes. However, it would be helpful to provide more context for these findings, such as why the sex ratio is important or what the implications of the protracted reproductive season are.

++ Introduction

The technical terms and concepts used in the introduction are appropriate for the target audience. However, some acronyms and terms could be defined for clarity (e.g., GSA 17, GSI, K Fulton).

The introduction is detailed but could be more concise. Some sentences are lengthy and can be broken down for better readability. For example, the description of the habitat and distribution of Squilla mantis could be simplified.

There are several typographical errors and formatting issues, such as inconsistent spacing (e.g., "S.mantis"). These should be corrected for a professional presentation.

Ensure consistent formatting throughout the text, such as italicizing scientific names and defining acronyms upon first mention. For example, "Squilla mantis" should be consistently italicized, and acronyms like GSA should be defined.

++ Materials and Methods

The multi-year sampling approach (2016-2020) is robust and provides a comprehensive dataset. The use of professional fishing fleets and adherence to the Data Collection Framework (DCF) ensures data reliability. However, the exclusion of August due to the fishing ban should be discussed in terms of its potential impact on the results.

The method for determining sex and macroscopic maturity stages is well-defined, but the grouping of "quiescent" and "spent" stages due to their similarity could introduce classification errors. Clarification on how this potential bias was mitigated would be useful.

he methods for measuring carapace length and total weight are standard and appropriate. The use of the length-weight relationship equation is scientifically sound. However, the description of the allometry coefficient could benefit from further explanation or a reference to a standard text for readers unfamiliar with the concept.

The use of specific equipment (e.g., callipers, light microscope, Leica Application Suite) and software (R version 4.0.0) is appropriate and standard for such studies. However, more details on the settings or configurations used during measurements and analysis could enhance reproducibility.

the statistical methods used (Chi-square test, t-test, ANOVA, logistic regression) are appropriate for the data and objectives. The use of R software and specific packages (e.g., sizeMat) is suitable for the analyses performed. However, the description of the non-linear regression analysis on log-log transformed data could be elaborated further for clarity.

The methods section is detailed and generally well-organized. However, some sentences are lengthy and could be broken down for better readability. For example, the description of the sex ratio estimation could be simplified.

"S.mantis" should be "S. mantis"

++Results

There are some minor grammatical errors and awkward phrasings that could be improved for better readability. For example, "significatively different" should be "significantly different."

The formatting of statistical results (e.g., p-values) should be consistent throughout the text. For instance, "p-value=0.05" should be formatted the same way as other p-values (e.g., "p<0.05").

++ Discussion

Lack of Novelty: The study reaffirms known trends without significant new findings. The sex ratio bias towards females and reproductive seasonality align with existing literature.

Limited Scope: The study focuses on a single geographical area, which limits the broader applicability of the findings.

Insufficient Context: The discussion lacks a deeper analysis of how these findings contribute to the management of the species and their implications on broader ecological or fisheries management practices.

Comparative Analysis: The comparison with other studies is brief and could be more comprehensive, discussing possible reasons for observed differences or similarities.

Clarity and Conciseness: Some sentences are overly complex and can be simplified for better readability.

Redundancy: Certain points are repeated, which could be condensed to make the discussion more succinct.

Grammar and Syntax: Some grammatical errors and awkward phrasing need correction.

Novelty Assessment: The study presents data specific to the Central Mediterranean but does not offer groundbreaking new insights. The validation of the macroscopic maturity scale with histological analysis is useful but not highly innovative.

++ Conclusions

The conclusion summarizes the findings but lacks detailed implications or recommendations for future research or fisheries management.

While it mentions the validation of the macroscopic maturity scale, it doesn't delve into the significance or limitations of this validation

The conclusion could benefit from more specific details about the methodologies and findings to reinforce the study's contributions.

The conclusion should connect the findings to broader ecological and management contexts more explicitly.

Novelty Assessment: The study's novelty lies in the histological validation of the macroscopic maturity scale for S. mantis in the Central Mediterranean, providing useful data for fisheries management. However, the conclusions do not emphasize this novelty strongly.

Revised Conclusions Conclusions

This study provided an updated overview of the reproductive pattern of S. mantis, contributing to the existing knowledge on its biology. For the first time, a macroscopic maturity scale was histologically validated to ensure an accurate evaluation of the reproductive cycle of this species in the Central Mediterranean Sea (GSA 17). Histological analysis identified the spot-tail mantis shrimp as a synchronous species, consistent with observations in other Mediterranean and Atlantic areas. The monthly distribution of maturity stages and the trend of the gonadosomatic index revealed a protracted spawning period, with significant peaks from March to May. These findings offer valuable insights for the correct management and sustainable exploitation of this important fishery resource.

Comments on the Quality of English Language

++Simple Summary:

Ensure consistent formatting of the species name, Squilla mantis, in italics throughout the article.

++ Abstract

The technical terms used (e.g., macroscopic maturity scale, histological analysis, somatic indexes, GSI, K Fulton) are suitable for the intended scientific audience. However, defining acronyms such as Gonadosomatic index (GSI) and Fulton’s Condition Factor (K) upon their first mention would be beneficial.

The abstract presents key findings, such as the sex ratio, length-weight relationship, and reproductive season, which are important for understanding the study's outcomes. However, it would be helpful to provide more context for these findings, such as why the sex ratio is important or what the implications of the protracted reproductive season are.

++ Introduction

The technical terms and concepts used in the introduction are appropriate for the target audience. However, some acronyms and terms could be defined for clarity (e.g., GSA 17, GSI, K Fulton).

The introduction is detailed but could be more concise. Some sentences are lengthy and can be broken down for better readability. For example, the description of the habitat and distribution of Squilla mantis could be simplified.

There are several typographical errors and formatting issues, such as inconsistent spacing (e.g., "S.mantis"). These should be corrected for a professional presentation.

Ensure consistent formatting throughout the text, such as italicizing scientific names and defining acronyms upon first mention. For example, "Squilla mantis" should be consistently italicized, and acronyms like GSA should be defined.

++ Materials and Methods

The multi-year sampling approach (2016-2020) is robust and provides a comprehensive dataset. The use of professional fishing fleets and adherence to the Data Collection Framework (DCF) ensures data reliability. However, the exclusion of August due to the fishing ban should be discussed in terms of its potential impact on the results.

The method for determining sex and macroscopic maturity stages is well-defined, but the grouping of "quiescent" and "spent" stages due to their similarity could introduce classification errors. Clarification on how this potential bias was mitigated would be useful.

he methods for measuring carapace length and total weight are standard and appropriate. The use of the length-weight relationship equation is scientifically sound. However, the description of the allometry coefficient could benefit from further explanation or a reference to a standard text for readers unfamiliar with the concept.

The use of specific equipment (e.g., callipers, light microscope, Leica Application Suite) and software (R version 4.0.0) is appropriate and standard for such studies. However, more details on the settings or configurations used during measurements and analysis could enhance reproducibility.

the statistical methods used (Chi-square test, t-test, ANOVA, logistic regression) are appropriate for the data and objectives. The use of R software and specific packages (e.g., sizeMat) is suitable for the analyses performed. However, the description of the non-linear regression analysis on log-log transformed data could be elaborated further for clarity.

The methods section is detailed and generally well-organized. However, some sentences are lengthy and could be broken down for better readability. For example, the description of the sex ratio estimation could be simplified.

"S.mantis" should be "S. mantis"

++Results

There are some minor grammatical errors and awkward phrasings that could be improved for better readability. For example, "significatively different" should be "significantly different."

The formatting of statistical results (e.g., p-values) should be consistent throughout the text. For instance, "p-value=0.05" should be formatted the same way as other p-values (e.g., "p<0.05").

++ Discussion

Lack of Novelty: The study reaffirms known trends without significant new findings. The sex ratio bias towards females and reproductive seasonality align with existing literature.

Limited Scope: The study focuses on a single geographical area, which limits the broader applicability of the findings.

Insufficient Context: The discussion lacks a deeper analysis of how these findings contribute to the management of the species and their implications on broader ecological or fisheries management practices.

Comparative Analysis: The comparison with other studies is brief and could be more comprehensive, discussing possible reasons for observed differences or similarities.

Clarity and Conciseness: Some sentences are overly complex and can be simplified for better readability.

Redundancy: Certain points are repeated, which could be condensed to make the discussion more succinct.

Grammar and Syntax: Some grammatical errors and awkward phrasing need correction.

Novelty Assessment: The study presents data specific to the Central Mediterranean but does not offer groundbreaking new insights. The validation of the macroscopic maturity scale with histological analysis is useful but not highly innovative.

++ Conclusions

The conclusion summarizes the findings but lacks detailed implications or recommendations for future research or fisheries management.

While it mentions the validation of the macroscopic maturity scale, it doesn't delve into the significance or limitations of this validation

The conclusion could benefit from more specific details about the methodologies and findings to reinforce the study's contributions.

The conclusion should connect the findings to broader ecological and management contexts more explicitly.

Novelty Assessment: The study's novelty lies in the histological validation of the macroscopic maturity scale for S. mantis in the Central Mediterranean, providing useful data for fisheries management. However, the conclusions do not emphasize this novelty strongly.

Revised Conclusions Conclusions

This study provided an updated overview of the reproductive pattern of S. mantis, contributing to the existing knowledge on its biology. For the first time, a macroscopic maturity scale was histologically validated to ensure an accurate evaluation of the reproductive cycle of this species in the Central Mediterranean Sea (GSA 17). Histological analysis identified the spot-tail mantis shrimp as a synchronous species, consistent with observations in other Mediterranean and Atlantic areas. The monthly distribution of maturity stages and the trend of the gonadosomatic index revealed a protracted spawning period, with significant peaks from March to May. These findings offer valuable insights for the correct management and sustainable exploitation of this important fishery resource.

Reviewer 3 Report

Comments and Suggestions for Authors

I enjoyed reading this well done study which extends and clarifies information on reproduction and populations dynamics in this commercially and ecologically important stomatopod species.  I have made comments and suggestions for polishing this fine contribution even further on the attached Word copy of the manuscript. I have made minor edits (highlighted in yellow) on wording and minor errors (e.g, not italicizing species names here and there, polishing the English just a bit). 

The presentation of sex ratio (the wording) is often confusing but need not be. Just use one method (females/males OR females/males plus females; I prefer the latter). The table on sex ration seems to blend them both or? Just make it clear please; also, in the Discussion, the authors could hypothesize a bit on the adaptive value on the specific seasonal pattern both for GSI and Condition Factor (see my suggestions there)

Comments on the Quality of English Language

On the attached Word copy, I have made some minor edits on wording and grammar (highlighted in yellow). A final good copy edit by journal staff would be good to do.

Round 2

Reviewer 2 Report

Comments and Suggestions for Authors

No comments.